Effect on speech emotion classification of a feature selection approach using a convolutional neural network

Amjad Ammar 1
http://orcid.org/0000-0002-1624-7580 Khan Lal 1
Chang Hsien-Tsung 1 2 3 4 smallpig@widelab.org
1 Department of Computer Science and Information Engineering, Chang Gung University , Taoyuan , Taiwan
2 Department of Physical Medicine and Rehabilitation, Chang Gung Memorial Hospital , Taoyuan , Taiwan
3 Artificial Intelligence Research Center, Chang Gung University , Taoyuan , Taiwan
4 Bachelor Program in Artificial Intelligence, Chang Gung University , Taoyuan , Taiwan
Shuja Junaid
Electronic publication date: 2021 Nov 3
Publication date: 2021
Volume: 7
Electronic Location ID: e766
Received 2021 Jul 30; Accepted 2021 Oct 11
Copyright: © 2021 Amjad et al.
Copyright year: 2021
Copyright holder: Amjad et al.
License: This is an open access article distributed under the terms of the Creative Commons Attribution License, which permits unrestricted use, distribution, reproduction and adaptation in any medium and for any purpose provided that it is properly attributed. For attribution, the original author(s), title, publication source (PeerJ Computer Science) and either DOI or URL of the article must be cited.
License URL: https://creativecommons.org/licenses/by/4.0/

Keywords: Speech emotion recognition, Feature extraction, Feature selection, Convolutional neural network, Mel-spectrogram, Data augmentation

Funding: Chang Gung Memorial Hospital CMRPD2J0023 Chang Gung University BMRPA07 This work was supported in part by Chang Gung Memorial Hospital under Grant CMRPD2J0023, and in part by Chang Gung University under Grant BMRPA07. The funders had no role in study design, data collection and analysis, decision to publish, or preparation of the manuscript.

==============================
Speech emotion recognition (SER) is a challenging issue because it is not clear which features are effective for classification. Emotionally related features are always extracted from speech signals for emotional classification. Handcrafted features are mainly used for emotional identification from audio signals. However, these features are not sufficient to correctly identify the emotional state of the speaker. The advantages of a deep convolutional neural network (DCNN) are investigated in the proposed work. A pretrained framework is used to extract the features from speech emotion databases. In this work, we adopt the feature selection (FS) approach to find the discriminative and most important features for SER. Many algorithms are used for the emotion classification problem. We use the random forest (RF), decision tree (DT), support vector machine (SVM), multilayer perceptron classifier (MLP), and k-nearest neighbors (KNN) to classify seven emotions. All experiments are performed by utilizing four different publicly accessible databases. Our method obtains accuracies of 92.02%, 88.77%, 93.61%, and 77.23% for Emo-DB, SAVEE, RAVDESS, and IEMOCAP, respectively, for speaker-dependent (SD) recognition with the feature selection method. Furthermore, compared to current handcrafted feature-based SER methods, the proposed method shows the best results for speaker-independent SER. For EMO-DB, all classifiers attain an accuracy of more than 80% with or without the feature selection technique.

Introduction

Recently, there has been much progress in artificial intelligence. However, we are still far short of interacting naturally with machines because machines can neither understand our emotional state nor our emotional behavior. In previous studies, some modalities have been proposed for identifying emotional states, such as extended text (Khan et al., 2021), speech (El Ayadi, Kamel & Karray, 2011), video (Hossain & Muhammad, 2019), facial expressions (Alreshidi & Ullah, 2020), short messages (Sailunaz et al., 2018), and physiological signals (Qing et al., 2019). These modalities vary across applications. The most common modalities in social media are emoticons and short text; video is the most common modality for gaming systems. Electroencephalogram signal-based emotion classification methods have also been introduced recently (Liu et al., 2020; Bazgir, Mohammadi & Habibi, 2018; Suhaimi, Mountstephens & Teo, 2020); however, the use of electroencephalogram signals is invasive and annoying for people.

Due to some inherent advantages, speech signals are the best source for affective computing. Speech signals can be obtained more economically and readily than other biological signals. Therefore, most researchers have focused on automatic speech emotion recognition (SER). There are numerous applications for identifying emotional persons, such as interactions with robots, entertainment, cardboard systems, commercial applications, computer games, audio surveillance, call centers, and banking.

Three main issues should be addressed to obtain a successful SER framework: (i) selecting an excellent emotional database, (ii) performing useful feature extraction, and (iii) using deep learning algorithms to design accurate classifiers. However, emotional feature extraction is a significant problem in an SER framework. In prior studies, many researchers have suggested significant features of speech, such as energy, intensity, pitch, standard deviation, cepstrum coefficients, Mel-frequency cepstrum coefficients (MFCCs), zero-crossing rate (ZCR), formant frequency, filter bank energy (FBR), linear prediction cepstrum coefficients (LPCCs), modulation spectral features (MSFs) and Mel-spectrograms. In Sezgin, Gunsel & Kurt (2012), several distinguishing acoustic features were used to identify emotions: spectral, qualitative, continuous, and Teager energy operator-based (TEO) features. Thus, many researchers have suggested that the feature set comprises more speech emotion information (Rayaluru, Bandela & Kishore Kumar, 2019). However, combining feature sets complicates the learning process and enhances the possibility of overfitting. In the last five years, researchers have presented many classification algorithms, such as the hidden Markov model (HMM) (Mao et al., 2019), support vector machine (SVM) (Kurpukdee et al., 2017), deep belief network (DBN) (Shi, 2018), K-nearest neighbors (KNN) (Zheng, Wang & Jia, 2020) and bidirectional long short-term memory networks (BiLSTMs) (Mustaqeem, Sajjad & Kwon, 2020). Some researchers have also suggested different classifiers; in the brain emotional learning model (BEL) (Mustaqeem, Sajjad & Kwon, 2020), a multilayer perceptron (MLP) and adaptive neuro-fuzzy inference system are combined for SER. The multikernel Gaussian process (GP) (Chen et al., 2016b) is another proposed classification strategy with two related notions. These provide for learning in the algorithm by combining two functions: the radial basis function (RBF) and the linear kernel function. In Chen et al. (2016b), the proposed system extracted two spectral features and used these two features to train different machine learning models. The proposed technique estimated that the combined features had high accuracy, above 90 percent on the Spanish emotional database and 80 percent on the Berlin emotional database. Han, Yu & Tashev (2014) adopted both utterance-and segment-level features to identify emotions.

Some researchers have weighted the advantages and disadvantages of each feature. However, no one has identified which feature is the best feature among feature categories (El Ayadi, Kamel & Karray, 2011; Sun, Wen & Wang, 2015; Anagnostopoulos, Iliou & Giannoukos, 2015). Many deep learning models have been proposed in SER to determine the high-level emotion features of utterances to establish a hierarchical representation of speech. The accuracy of handcrafted features is relatively high, and this feature extraction technique always requires manual labor (Anagnostopoulos, Iliou & Giannoukos, 2015; Chen, Mao & Yan, 2016a, Chen et al., 2012). The extraction of handcrafted features usually ignores the high-level features. However, the best and most appropriate features that are emotionally powerful must be selected by effective performance for SER.

Therefore, it is more important to select specific speech features that are not affected by country, speaking style of the speaker, culture, or region. Feature selection (FS) is also essential after extraction and is accompanied by an appropriate classifier to recognize emotions from speech. A summary of FS is presented in Kerkeni et al. (2019). Both feature extraction and FS effectively reduce computational complexity, enhance learning effectiveness, and reduce the storage needed. To extract the local features, we use a convolutional neural network (CNN) (AlexNet). The CNN automatically extracts the appropriate local features from the augmented input spectrogram of an audio speech signal. When using CNNs for the SER system, the spectrogram is frequently used as the CNN input to obtain high-level features. In recent years, numerous studies have been presented, such as (Abdel-Hamid et al., 2014; Krizhevsky, Sutskever & Hinton, 2017). The authors used a CNN model for feature extraction of audio speech signals. Recently, deep learning models such as AlexNet (Li et al., 2021), VGG (Simonyan & Zisserman, 2015), and ResNet (He et al., 2015) have been used extensively to perform different classification tasks. Additionally, these deep learning models regularly perform much better than shallow CNNs. The main reason is that deep CNNs extract mid-level features from the input data using multilevel convolutional and pooling layers. The detailed abbreviations and definitions used in the paper are listed in Table 1.

Table 1 Nomenclature.

ACRNN	Attention convolutional recurrent neural network	KNN	K-nearest neighbors	
BEL	Brain emotional learning	LPCC	Linear predictive cepstral coefficients	
BiLSTM	Bidirectional long short-term memory	MFCC	Mel frequency cepstral coefficients	
CNN	Convolutional neural network	MLP	Multilayer perceptron	
CL	Convolutional layer	MSF	Modulation spectral features	
CNN	Convolutional neural network	PAD	Pleasure-arousal-dominance	
CFS	Correlation-based feature selection	PL	Pooling layer	
DBN	Deep belief network	RBFNN	Radial basis function neural network	
DCNN	Deep convolutional neural network	RBF	Radial basis function	
DNN	Deep neural network	RF	Random forest	
DRCNN	Deep retinal CNNs	RP	Residual phase	
DT	Decision tree	RNN	Recurrent neural network	
FS	Feature selection	SAVEE	Surrey audio-visual expressed emotion	
FCL	Fully connected layer	SD	Speaker-dependent	
FBR	Filter bank energy	SI	Speaker-independent	
GMM	Gaussian mixture model	SVM	Support vector machine	
GP	Gaussian process	SER	Speech emotion recognition	
HMM	Hidden markov model	TEO	Teager energy operator	
KELM	Kernel extreme learning machine	ZCR	Zero-crossing rate	

The main contributions of this paper are as follows: (1) In the proposed study, AlexNet is used to extract features for a speech emotion recognition system. (2) A feature selection approach is used to enhance the accuracy of SER. (3) The proposed approach performs better than existing handcrafted and deep-learning methods for SD and SI experiments.

The rest of the paper is organized as follows: Part 2 reviews the previous work in SER related to this paper’s current study. A detailed description of the emotional dataset used in the presented work and the proposed method for FS and the classifier are discussed in Part 3. The results are discussed in Part 4. Part 5 contains the conclusion and outlines future work.

Background

In this study, five different machine learning algorithms are used for emotion recognition tasks. There are two main parts of SER. One part is based on distinguishing feature extraction from audio signals. The second part is based on selecting a classifier that classifies emotional classes from speech utterances.

Speech emotion recognition using machine learning approaches

Researchers have used different machine learning classifiers to identify emotional classes from speech: SVM (Sezgin, Gunsel & Kurt, 2012), random forest (RF) (Noroozi et al., 2017), Gaussian mixture models (GMMs) (Patel et al., 2017), HMMs (Mao et al., 2019), CNNs (Christy et al., 2020), k-nearest neighbors (KNN) (Kapoor & Thakur, 2021) and MLP. These algorithms have been commonly used to identify emotions. Emotions are categorized using two approaches: categorical and dimensional approaches. Emotions are classified into small groups in the categorical approach. Ekman (1992) proposed six basic emotions: anger, happiness, sadness, fear, surprise, and disgust. In the second category, emotions are defined by axes with a combination of several dimensions (Costanzi et al., 2019). Different researchers have described emotions relative to one or more dimensions. Pleasure-arousal-dominance (PAD) is a three-dimensional emotional state model proposed by Mehrabian (1996). Different features are essential in identifying speech emotions from voice. Spectral features are significant and widely used to classify emotions. A decision tree was used to identify emotions from the CASIA Chinese emotion corpus in Tao et al. (2008) and achieved 89.6% accuracy. Kandali, Routray & Basu (2009) introduced an approach to classify emotion-founded MFCCs as the main features and applied a GMM as a classifier. Milton, Roy & Selvi (2013) presented a three-stage traditional SVM classifying different Berlin emotional datasets. Waghmare et al. (2014) adopted spectral features (MFCCs) as the main feature and classified emotions from the Marathi speech dataset. Demircan & Kahramanli (2014) extracted MFCC features from the Berlin EmoDB database. They used the KNN algorithm to recognize speech emotions. The Berlin emotional speech database (EMO-DB) was used in the experiment, and the accuracy obtained was between 90% and 99.5%. Hossain & Shamim (2014) proposed a cloud-based collaborative media system that uses emotions from speech signals and uses standard features such as MFCCs. Paralinguistic features and prosodic features were utilized to detect emotions from speech in Alonso et al. (2015). SVM, a radial basis function neural network (RBFNN), and an autoassociative neural network (AANN) were used to recognize emotions after combining two features, MFCCs and the residual phase (RP), from a music database (Nalini & Palanivel, 2016). SVMs and DBNs were examined utilizing the Chinese academic database (Zhang et al., 2017). The accuracy using DBNs was 94.5%, and the accuracy of the SVM was approximately 85%. In Yogesh et al. (2017), particle swarm optimization-based features and high-order statistical features were utilized. Chourasia et al. (2021) implemented an SVM and HMM to classify speech emotions after extracting the spectral features from speech signals.

Speech emotion recognition using deep learning approaches

Low-level handcrafted features are very useful in distinguishing speech emotions. With many successful deep neural network (DNN) applications, many experts have started to target in-depth emotional feature learning. Schmidt & Kim (2011) used an approach based on linear regression and deep belief networks to identify musical emotions. They used the MoodSwings Lite music database and obtained a 5.41% error rate. Le & Provost (2013) implemented hybrid classifiers, which were a set of DBNs and HMMs, and attained good results on FAU Aibo. Deng et al. (2013) presented a transfer learning feature method for speech emotion recognition based on a sparse autoencoder. Several databases were used, including the eNTERFACE and EMO-DB databases (Deng et al., 2013). In Poon-Feng et al. (2014), a generalized discriminant analysis method (Gerda) was presented with several Boltzmann machines to analyze and classify emotions from speech and improve the previous reported baseline by traditional approaches. Schmidt & Kim (2011) proposed a regression-based DBN to recognize music emotions and a model based on three hidden layers to learn emotional features (Han, Yu & Tashev, 2014).

Trentin, Scherer & Schwenker (2015) proposed a probabilistic echo-state network-based emotion recognition framework that obtained an accuracy of 96.69% using the WaSep database. More recent work introduced deep retinal CNNs (DRCNNs) in Niu et al. (2017), which showed good performance in recognizing emotions from speech signals. The presented approach obtained the highest accuracy, 99.25%, in the IEMOCAP database. In Fayek, Lech & Cavedon (2017), the authors suggested deep learning approaches. A speech signal spectrogram was used as an input. The signal may be represented in terms of time and frequency. The spectrogram is a fundamental and efficient way to describe emotional speech impulses in the time-frequency domain. It has been used with particular effectiveness for voice and speaker recognition and word recognition (Stolar et al., 2017). In Stolar et al. (2017), the existing approach used ALEXNet-SVM, experiments were performed on the EMO-DB database with seven emotions. Satt, Rozenberg & Hoory (2017) suggested another efficient convolutional LSTM approach for emotion classification. The introduced model learned spatial patterns and spatial spectrogram patterns representing information on the emotional states. The experiment was performed on the IEMOCAP database with four emotions. Two different databases were used to extract prosodic and spectral features with an ensemble softmax regression approach (Sun & Wen, 2017). For the identification of emotional groups, experiments were performed on the two different datasets. A CNN was used in Fayek, Lech & Cavedon (2017) to classify four emotions from the IEMOCAP database: happy, neutral, angry, and sad. In Xia & Liu (2017), multitasking learning was used to obtain activation and valence data for speech emotion detection using the DBN model. IEMOCAP was used in the experiment to identify the four emotions. However, high computational costs and a large amount of data are required for deep learning techniques. The majority of current speech emotional databases have a small amount of data. Deep learning model approaches are insufficient for training with large-scale parameters. A pretrained deep learning model is used based on the above studies. In Badshah et al. (2017), a pretrained DCNN model was introduced for speech emotion recognition. The outcomes were improved with seven emotional states. In Badshah et al. (2017), The authors suggested a DCNN accompanied by a discriminant temporal pyramid matching with four different databases. In the suggested approach, the authors used six emotional classes for BAUM-1s, eNTERFACE05, RML databases and used seven emotions for the Emo-DB databases. DNNs were used to divide emotional probabilities into segments (Gu, Chen & Marsic, 2018), which were utilized to create utterance features; these probabilities were fed to the classifier. The IEMOCAP database was used in the experiment, and the obtained accuracy was 54.3%. In Zhao et al. (2018), the suggested approach used integrated attention with a fully convolutional network (FCN) to automatically learn the optimal spatiotemporal representations of signals from the IEMOCAP database. The hybrid architecture proposed in Etienne et al. (2018) included a data augmentation technique. In Wang & Guan (2008) and Zhang et al. (2018), the fully connected layer (FC7) of AlexNet was used for the extraction process. The results were evaluated on four different databases with six emotional states. In Guo et al. (2018), an approach for SER that combined phase and amplitude information utilizing a CNN was investigated. In Chen et al. (2018), a three-dimensional convolutional recurrent neural network including an attention mechanism (ACRNN) was introduced. The identification of emotion was evaluated using the Emo-DB and IEMOCAP databases. The attention process was used to develop a dilated CNN and BiLSTM in Meng et al. (2019). To identify speech emotion, 3D log-Mel spectrograms were examined for global contextual statistics and local correlations. The OpenSMILE package was used to extract features in Özseven (2019). The accuracy obtained with the Emo-DB database was 84%, and it was 72% with the SAVEE database. Pretrained networks have many benefits, including the ability to reduce the training time and improve accuracy. Kernel extreme learning machine (KELM) features were introduced in Guo et al. (2019). An adversarial data augmentation network was presented in Yi & Mak (2019) to create simulated samples to resolve the data scarcity problem. Energy and pitch were extracted from each audio segment in Ververidis & Kotropoulos (2005), Rao, Koolagudi & Vempada (2013), and Daneshfar, Kabudian & Neekabadi (2020). They also needed fewer training data and could deal directly with dynamic variables. Two different acoustic paralinguistic feature sets were used in Haider et al. (2021). An implementation of real-time voice emotion identification using AlexNet was described in Lech et al. (2020). When trained on the Berlin Emotional Speech (EMO-DB) database with six emotional classes, the presented method obtained an average accuracy of 82%. According to existing research (Stolar et al., 2017; Badshah et al., 2017; Lech et al., 2020) most of the studies used simulated databases with few emotional states. On the other hand, in the proposed study, we utilized eight emotional states for RAVDESS, seven emotional states for SAVEE, six emotional states for Emo-DB, and four emotional states for the IEMOC AP database. Therefore, our results are state-of-the-art for simulated and semi-natural databases.

Proposed method

This section describes the proposed pretrained CNN (AlexNet) algorithm for the SER framework. We fine-tune the pretrained model (Krizhevsky, Sutskever & Hinton, 2017) on the created image-like Mel-spectrogram segments. We do not train our own deep CNN framework owing to the limited emotional audio dataset. Furthermore, computer vision experiments (Ren et al., 2017; Campos, Jou & i Nieto, 2017) have depicted that fine-tuning the pretrained CNNs on target data is acceptable to relieve the issue of data insufficiency. AlexNet is a model pretrained on the extensive ImageNet dataset, containing a wide range of different labeled classes, and uses a shorter training time. AlexNet (Krizhevsky, Sutskever & Hinton, 2017; Stolar et al., 2017; Lech et al., 2020) comprises five convolution layers, three max-pooling layers, and three fully connected layers. In the proposed work, we extract the low-level features from the fourth convolutional layer (CL4).

The architecture of our proposed model is displayed in Fig. 1. Our model comprises four processes: (a) development of the audio input data, (b) low-level feature extraction using AlexNet, (c) feature selection, and (d) classification. Below, we explain all four steps of our model in detail.

Figure 1 The structure of our proposed model for audio emotion recognition.

Creation of the audio input

In the proposed method, the Mel-spectrogram segment is generated from the original speech signal. We create three channels of the segment from the original 1D audio speech dataset. Then, the generated segments are converted into fixed-size 227 × 227 × 3 inputs for the proposed model. Following (Zhang et al., 2018), 64 Mel-filter banks are used to create the log Mel-spectrogram, and each frame is multiplied by a 25 ms window size with a 10 ms overlap. Then, we divide the log Mel spectrogram into fixed segments by using a 64-frame context window. Finally, after extracting the static segment, we calculate the regression coefficients of the first and second order around the time axis, thereby generating the delta and double-delta coefficients of the static Mel spectrogram segment. Consequently, three channels with 64 × 64 × 3 Mel-spectrogram segments can be generated as the inputs of AlexNet, and these channels are identical to the color RGB image. Therefore, we resize the original 64 × 64 × 3 spectrogram to the new size 227 × 227 × 3. In this case, we can create four (middle, side, left, and right) segments of the Mel spectrogram, as shown in Fig. 2.

Figure 2 The general architecture of AlexNet, The parameters of the convolutional layer are represented by the “Conv(kernel size)-[stride size]-[number of channels]”.

The parameters of themax-pooling layer are indicated as “Maxpool-[kernel size]-[stride size]”.

Emotion recognition using AlexNet

In the proposed method, CL4 of the pretrained model is used for feature extraction. The CFS feature selection approach is used to select the most discriminative features. The CFS approach selects only very highly correlated features with output class labels. The five different classification models are used to test the accuracy of the feature subsets.

Feature extraction

In this study, feature extraction is performed using a pretrained model. The original weight of the model remains fixed, and existing layers are used to extract the features. The pretrained model has a deep structure that contains extra filters for every layer and stacked CLs. It also includes convolutional layers, max-pooling layers, momentum stochastic gradient descent, activation functions, data augmentation, and dropout. AlexNet uses a rectified linear unit (ReLU) activation function. The layers of the network are explained below.

Input layer

This layer of the pretrained model is a fixed-size input layer. We resample the Mel spectrogram of the signal to a fixed size 227 × 227 × 3.

Convolutional layer (CL)

The convolutional layer is composed of convolutional filters. Convolutional filters are used to obtain many local features in the input data from local regions to form various feature groups. AlexNet contains five CLs, in which three layers follow the max-pooling layer. CL1 includes 96 kernels with a size of 11 × 11 × 3, zero padding, and a stride of four pixels. CL2 contains 256 kernels, each of which is 5 × 5 × 48 in size and includes a one-pixel stride and a padding value of 2. The CL3 contains 384 kernels of size 3 × 3 × 256. CL4 contains 384 kernels of size 3 × 3 × 192. For the output value of each CL, the ReLU function is used, which speeds up the training process.

Pooling layer (PL)

After the CLs, a pooling layer is used. The goal of the pooling layer is to subsample the feature groups. The feature groups are obtained from the previous CLs to create a single data convolutional feature group from the local areas. Average pooling and max-pooling are the two basic pooling operations. The max-pooling layer employs maximum filter activation across different points in a quantified frame to produce a modified resolution type of CL activation.

Fully connected layers (FCLs)

Fully connected layers incorporate the characteristics acquired from the PL and create a feature vector for classification. The output of the CLs and PLs is given to the fully connected layers. There are three fully connected layers in AlexNet: FC6, FC7, and FC8. A 4,096-dimensional feature map is generated by FC6 and FC7, while FC8 generates 1,000-dimensional feature groups.

Feature maps can be created using FCLs. These are universal approximations, but fully connected layers do not work fully in recognizing and generalizing the original image pixels. CL4 extracts relevant features from the original pixel values by preserving the spatial correlations inside the image. Consequently, in the experimental setup, features are extracted from the CL4 employed for SER. A total of 64,896 features are obtained from CL4. Certain features are followed by a FS method and pass through a classification model for identification. Table 2(a) represents a detailed layers architecture of proposed model. AlexNet required 227 × 227 size RGB images as input. Each convolution filter yields a stack of the feature map. The learning approach starts with an initial learning rate of 0.001 and gradually decreases with a drop rate of 0.1. By using 96 filters of 11 × 11 × 3, CL1 creates an array of activation maps. As a consequence, CL4 generates 384 activation maps (3 × 3 × 192 filters).

Table 2 (a) Alexnet layers architecture and (b) number of selected features after CFS.

(a) Layer type	Size	Kernels size	Number of features	
Image input	227 × 227 × 3		150,528	
Convolution layer#1	11 × 11 × 3	96	253,440	
Activation function				
Channel normalization				
Pooling				
Convolution layer#2	5 × 5 × 48	256	186,624	
Activation function				
Convolution layer#3	3 × 3 × 256	384	64,896	
Activation function				
Channel normalization				
Pooling				
Convolution layer#4	3 × 3 × 192	384	64,896	
Activation function				
Convolution layer#5	3 × 3 × 192	256	43,264	
Activation function				
Pooling				
Fully connected layer			4,096	
Activation function				
Dropout				
Fully connected layer			4,096	
Activation function				
Dropout				
Fully connected layer			1,000	
(b) Database	Number of extracted features	No. of best features using CFS	
Emo-DB	64,896	458	
SAVEE	64,896	150	
IEMOCAP	64,896	445	
RAVDESS	64,896	267	

Feature selection

The discriminative and related features for the model are determined by feature selection. FS approaches are used with several models to minimize the training time and enhance the ability to generalize by decreasing overfitting. The main goal of feature selection is to remove insignificant and redundant features.

Correlation-based measure

We can identify an excellent feature if it is related to the class features and is not redundant with respect to any other class features. For this reason, we use entropy-based information theory. The equation of entropy-based information theory is defined as:

(1) F(E)=−ΣS(ej)log2(S(ej)).

The entropy of E after examining the values of G is defined in the equation below:

(2) F(E/G)=−ΣS(gk)ΣS(ej/gk)log2(S(ej/gk))

S(ej) denotes the probability for all values of E, whereas S(ej/gk) denotes the probabilities of E when the values of G are specified. The percentage by which the entropy of E decreases reflects the irrelevant information about E given by G, which is known as information gain. The equation of information gain is given below:

(3) I(E/G)=(F(E)−F(E/G)).

If I(E/G) > I(H/G), then we can conclude that feature G is much more closely correlated to feature E than to feature H. We possess one more metric, symmetrical uncertainty, which indicates the correlation between features, defined by the equation below:

(4) SU(E,G)=2[I(E/G)/F(E)+F(G)].

SU balances the information gain bias toward features with more values by normalizing its value to the range [0, 1]. SU analyzes a pair of features symmetrically. Entropy-based techniques need nominal features. These features can be used to evaluate the correlations between continuous features if these features are discretized properly.

We use the correlation feature-based approach (CFS) (Wosiak & Zakrzewska, 2018) in the proposed work based on the previously described techniques. It evaluates a subset of features and selects only highly correlated discriminative attributes. CFS ranks the features by applying a heuristic correlation evaluation function. It estimates the correlation within the features. CFS drops unrelated features that have limited similarity with the class label. The CFS equation is as follows:

(5) FS=maxSkrcf1+rcf2+rcf3+....+rcfkk+2(rf1f2+....+rfifj+....+rfkfk−1),

where k represents the total number of features, rcfi represents the classification correlation of the features, and rfifj represents the correlation between features. The extracted features are fed into classification algorithms. CFS usually deletes (backward selection) or adds (forward selection) one feature at a time. Table 2(b) gives the most discriminative number of selected features.

Classification methods

The discriminative features provide input to the classifiers for emotion classification. In the proposed method, five different classifiers, KNN, RF, decision tree, MLP, and SVM, are used to evaluate the performance of speech emotion recognition.

Support vector machine (SVM)

SVMs are used for binary regression and classification. They create an optimal higher-dimensional space with a maximum class margin. SVMs identify the support vectors vj, weights wfj, and bias b to categorize the input information. For classification of the data, the following expression is used:

(6) sk(v,vj)=(ρvevj+k)z.

In the above equations, k is a constant value, and b represents the degree of the polynomial. For a polynomial ρ > zero:

(7) v=(Σi=0nwfjsk(vj,v)+b.

In the above equation, sk represents the kernel function, v is the input, vj is the support vector, wfj is the weight, and b is the bias. In our study, we utilize the polynomial kernel to translate the data into a higher-dimensional space.

K-nearest neighbors (KNN)

This classification algorithm keeps all data elements. It identifies the most comparable N examples and employs the target class emotions for all data examples based on similarity measures. In the proposed study, we fixed N = 10 for emotional classification. The KNN method finds the ten closest neighbors using the Euclidean distance, and emotional identification is performed using a majority vote.

Random forest (RF)

An RF is a classification and regression ensemble learning classifier. It creates a class of decision trees and a meaningful indicator of the individual trees for data training. The RF replaces each tree in the database at random, resulting in unique trees, in a process called bagging. The RF splits classification networks based on an arbitrary subset of characteristics per tree.

Multilayer perceptron (MLP)

MLPs are neural networks that are widely employed in feedforward processes. They consist of multiple computational levels. Identification issues may be solved using MLPs. They use a supervised back-propagation method for classifying occurrences. The MLP classification model consists of three layers: the input layer, the hidden layers, and the output layer. The input layer contains neurons that are directly proportional to the features. The degree of the hidden layers depends on the overall degree of the emotions in the database. It features dimensions after the feature selection approach. The number of output neurons in the database is equivalent to the number of emotions. The sigmoid activation function utilized in this study is represented as follows:

(8) pi=11+e−qi

In the above equation, the state is represented by pi, whereas the entire weighted input is represented by qi. When using the Emo-DB database, there is only one hidden layer in the MLP. It has 232 neurons. When using the SAVEE database, there is only one hidden layer in the MLP, and it comprises 90 neurons. The MLP contains a single hidden layer, and 140 neurons are present in the IEMOCAP database. In comparison, one hidden layer and 285 neurons are present in the RAVDESS dataset. The MLP is a two-level architecture; thus, identification requires two levels: training and testing. The weight values are set throughout the training phase to match them to the particular output class.

Experiments

Datasets

This experimental study contains four emotional speech databases, and these databases are publicly available, represented in Table 3(a).

• Ryerson Audio-Visual Database of Emotional Speech and Song (RAVDESS): RAVDESS is an audio and video database consisting of eight acted emotional categories: calm, neutral, angry, surprise, fear, happy, sad, and disgust, and these emotions are recorded only in North American English. RAVDESS was recorded by 12 male and 12 female professional actors.

• Surrey Audio-Visual Expressed Emotion (SAVEE): The SAVEE database contains 480 emotional utterances. The SAVEE database was recorded in British English by four male professional actors with seven emotion categories: sadness, neutral, frustration, happiness, disgust, anger, and surprise.

• Berlin Emotional Speech Database (Emo-DB): The Emo-DB dataset contains 535 utterances with seven emotion categories: neutral, fear, boredom, disgust, sad, angry, and joy. The Emo-DB emotional dataset was recorded in German by five male and five female native-speaker actors.

• Interactive Emotional Dyadic Motion Capture (IEMOCAP): The IEMOCAP multispeaker database contains approximately 12 hours of audio and video data with seven emotional states, surprise, happiness, sadness, anger, fear, excitement, and frustration, as well as neutral and other states. The IEMOCAP database was recorded by five male and five female professional actors. In this work, we use four (neutral, angry, sadness, and happiness) class labels. Table 3(b) illustrates the features of databases, which are used in a proposed method.

Table 3 (a) Detailed description of the datasets, (b) categories of emotional speech databases, theirfeatures, and some examples of each category.

(a) Datasets	Speakers	Emotions	Languages	Size	
RAVDESS	24 actors (12 male, 12 female)	Eight emotions (calm, neutral, angry, happy, fear, surprise, sad, disgust)	North American English	7,356 files (total size: 24.8 GB).	
SAVEE	4 (male)	Seven emotions (sadness, neutral, frustration, happiness, disgust,anger, surprise)	British English	480 utterances (120 utterances per speaker)	
Emo-DB	10 (5 male, 5 female)	Seven emotions (neutral, fear, boredom, disgust, sad, angry, joy)	German	535 utterances	
IEMOCAP	10 (5 male, 5 female)	Nine emotions (surprise, happiness, sadness, anger, fear, excitement, neutral, frustration and others)	English	12 hours of recordings	
(b)	Simulated	Semi natural	
Description	Generated by trained and experienced actors delivering the same sentence with different degrees of emotion	Created by having individuals read a script with a different emotions	
Single emotion at a time	Yes	Yes	
Widely used	Yes	No	
Copyrights and privacy protection	Yes	Yes	
Includes contextual information	No	Yes	
Includes situational information	No	Yes	
Emotions that are separate and distinct	Yes	No	
Numerous emotions	Yes	Yes	
Simple to model	Yes	No	
Numerous emotions	Yes	Yes	
Examples	EMO-DB,SAVEE, RAVDESS	IEMOCAP	

Experimental setup

All the experiments are completed in version 3.9.0 of the Python language framework. Numerous API libraries are used to train the five distinct models. The framework uses Ubuntu 20.04. The key objective is to implement an input data augmentation and feature selection approach for the five different models. The feature extraction technique is also involved in the proposed method. The lightweight and most straightforward model presented in the proposed study has excellent accuracy. In addition, low-cost complexity can monitor real-time speech emotion recognition systems and show the ability for real-time applications.

Anaconda

Anaconda is the best data processing and scientific computing platform for Python. It already includes numerous data science and machine learning libraries. Anaconda also includes many popular visualization libraries, such as matplotlib. It also provides the ability to build a different environment with a few unique libraries to carry out the task.

Keras

The implementation of our model for all four datasets was completed from scratch using Keras. It makes it extremely simple for the user to add and remove layers and activate and utilize the max-pooling layer in the network.

Librosa

Librosa (McFee et al., 2015) is a basic Python library used for this research. Librosa is used to examine the audio signal recordings. The four (side, middle, left, and right) segments of the Mel spectrogram were obtained through Librosa.

Experimental results and analysis

(Chau & Phung, 2013).

Speaker-dependent (SD) experiments

The performance of the proposed SER system is assessed using benchmark databases for the SD experiments. We use ten-fold cross-validation in our studies. All databases are divided randomly into ten equal complementary subsets with a dividing ratio of 80:20 to train and test the model. Table 4 gives the results achieved by five different classifiers utilizing the features extracted from CL4 of the model. The SVM achieved 92.11%, 87.65%, 82.98%, and 79.66% accuracies for the Emo-DB, RAVDESS, SAVEE and IEMODB databases, respectively. The proposed method reported the highest accuracy of 86.56% on the Emo-DB database with KNN. The MLP classifier obtained 86.75% accuracy for the IEMOCAP database. In contrast, the SVM reported 79.66% accuracy for the IEMOCAP database. The MLP classifier reported the highest accuracy, 91.51%, on the Emo-DB database. The RF attained 82.47% accuracy on the Emo-DB database, while DT achieved 80.53% accuracy on Emo-DB.

Table 4 Standard deviation and weighted average recall of the SD experiments without FS.

	SVM	RF	KNN	MLP	DT	
RAVDESS	87.65 ± 1.79	78.65 ± 4.94	78.15 ± 3.39	80.67 ± 2.89	76.28 ± 3.24	
SAVEE	82.98 ± 4.87	78.38 ± 4.10	79.81 ± 4.05	81.13 ± 3.63	69.15 ± 2.85	
Emo-DB	92.11± 2.29	82. 47± 3.52	86.56 ± 2.78	91.51 ± 2.09	80.53 ± 4.72	
IEMODB	79.66 ± 4.44	80.93 ± 3.75	74.33 ± 3.37	86.75 ± 3.64	67.25 ± 2.33	

Table 5 represents the results of the FS approach. The proposed FS technique selected 458 distinguishing features out of a total of 64,896 features for the Emo-DB dataset. The FS method obtained 150,445,267 feature maps for the SAVEE, RAVDESS, and IEMOCAP datasets.

Table 5 Standard deviation and weighted average recall of the SD experiments with FS.

Database	SVM	RF	KNN	MLP	DT	
RAVDESS	93.61 ± 1.32	85.21 ± 3.55	88.34 ± 2.67	84.50 ± 2.23	78.45 ± 2.67	
SAVEE	88.77 ± 2.45	86.79 ± 2.96	83.45 ± 3.21	85.45 ± 3.12	75.68 ± 3.82	
Emo-DB	96.02± 1.07	93. 51± 2.21	92.45 ± 2.45	95.80 ± 2.34	79.13 ± 4.01	
IEMODB	77.23 ± 2.66	86.23± 2.54	82.78 ± 2.17	89.12 ± 2.57	72.32 ± 1.72	

The experimental results illustrate a significant accuracy improvement by using data resampling and the FS approach. We consider the standard deviation and average weighted recall to evaluate the performance and stability of the SD experiments using the FS approach. The SVM classifier reached 93.61% and 96.02% accuracy for RAVDESS and Emo-DB, respectively, while the obtained accuracies were 88.77% and 77.23% for SAVEE and IEMOCAP, respectively, through the SVM. The MLP classifier obtained 95.80% and 89.12% accuracies with the Emo-DB and IEMOCAB databases, respectively.

The KNN classifier obtained the highest accuracy, 92.45% and 88.34%, with the Emo-DB and RAVDEES datasets. The RF classifier reported the highest accuracy, 93.51%, on the Emo-DB dataset and 86.79% accuracy on the SAVEE dataset with the feature selection approach. Table 5 shows that the SVM obtained better recognition accuracy than the other classification models with the FS method. A confusion matrix is an approach for describing the accuracy of the classification technique. For instance, if the data contains an imbalanced amount of samples in every group or more than two groups, the accuracy of the classification alone may be deceptive. Thus, calculating a confusion matrix provides a clearer understanding of what our classification model gets right and what kinds of mistakes it makes. It is common used in related researches (Zhang et al., 2018; Chen et al., 2018; Zhang, Zhao & Tian, 2019). The row means the actual emotion classes in the confusion matrix, while the column indicates the predicted emotion classes. The results of the confusion matrix are used to evaluate the identification accuracy of the individual emotional labels. The Emo-DB database contains seven emotional categories, three of which, “sad”, “disgust”, and “neutral,” were identified with accuracies of 98.88%, 98.78%, and 97.45%, respectively, by the SVM illustrated in Fig. 3. As shown in Fig. 4, the SVM recognized “frustration” and “neutral” with the highest accuracies, 97.78% and 92.45%, with the SAVEE dataset. As shown in Fig. 5, the RAVDESS dataset contains eight emotions, including “anger”, “calm”, “fear”, and “neutral”, which are listed with accuracies of 96.32%, 97.65%, 95.54%, and 99.98%, respectively. The IEMOCAP database identified "anger" with the highest accuracy of 93.23%, while “happy,” “sad,” and “neutral” were recognized with the highest accuracies of 83.41%, 91.45%, and 89.65% with the MLP classifier illustrated in Fig. 6, respectively.

Figure 3 Confusion matrix obtained by the SVM on the Emo-DB database for the SD experiment.

Figure 4 Confusion matrix obtained by the SVM on the SAVEE database for the SD experiment.

Figure 5 Confusion matrix obtained by the SVM on the RAVDESS database for the SD experiment.

Figure 6 Confusion matrix obtained by the MLP on the IEMOCAP database for the SD experiment.

Speaker-independent (SI) experiments

We adopted the single-speaker-out (SSO) method for the SI experiments. One annotator was used for testing, and all other annotators were used for training. In the proposed approach, the IEMOCAP dataset was split into testing and training sessions. By switching all of the testing annotators, the process was repeated, and the average accuracy was obtained for every testing speaker. Table 6 lists the identification results of five classification models for the SI experiments without the FS technique. The MLP obtained the highest accuracy, 88.32%, with the Emo-DB dataset. With the SAVE database, MLP obtained the highest accuracy, 65.18%. The SVM achieved the highest accuracy of 87.65% with Emo-DB and 75.34% with the RAVDESS database. The random forest achieved the highest accuracies, 79.45% and 65.78%, with Emo-DB and RAVDESS, respectively. Table 6 shows that the SVM obtained better recognition accuracy than the other classification models without the FS method. Table 7 represents the outcomes for the SI experiments with the feature extraction approach with data resampling and the FS method. The FS and data resampling approach improved the accuracy, according to the preliminary results.

Table 6 Standard deviation and weighted average recall of the SI experiment results without FS.

	SVM	RF	KNN	MLP	DT	
RAVDESS	75.34 ± 2.58	65.78 ± 2.32	69.12 ± 2.20	71.01 ± 2.84	67.41 ± 2.37	
SAVEE	63.02 ± 3.21	59.66 ± 3.79	71.81 ± 3.81	65.18 ± 2.05	59.55 ± 2.23	
Emo-DB	87.65± 2.56	79. 45± 2.11	75.30 ± 2.19	88.32 ± 2.67	76.27 ± 2.35	
IEMODB	61.85 ± 3.20	60.11 ± 4.20	55.47 ± 2.96	63.18 ± 1.62	54.69 ± 3.72	

Table 7 Standard deviation and weighted average recall of the SI experiment results with FS.

Database	SVM	RF	KNN	MLP	DT	
RAVDESS	80.94 ± 2.17	76.82 ± 2.16	75.57 ± 3.29	82.75 ± 2.10	76.18 ± 1.33	
SAVEE	70.06 ± 3.33	65.55 ± 2.42	60.58 ± 3.84	75.38 ± 2.74	63.69 ± 2.22	
Emo-DB	90.78± 2.45	85.73± 2.58	81.32 ± 2.12	92.65 ± 3.09	78.21 ± 3.47	
IEMODB	84.00 ± 2.76	78.08± 2.65	76.44 ± 3.88	80.23 ± 2.77	75.78 ± 2.25	

We report the average weighted recall and standard deviation to evaluate the SI experiment’s performance and stability utilizing the FS method. The SVM obtained the highest accuracies, 90.78%, 84.00%, 80.94%, and 70.06%, for the Emo-DB, IEMOCAP, RAVDESS, and SAVEE databases, respectively, followed by the FS method in the SI experiments. However, the MLP achieved the highest accuracies, 92.65%, 80.23%, 82.75%, and 75.38%, for the Emo-DB, IEMOCAP, RAVDESS, and SAVEE databases, respectively, followed by the FS method in the SI experiments. The confusion matrices of the results obtained for the SI experiments are shown in Figs. 7–9 to analyze the individual emotional groups’ identification accuracies. The average accuracies achieved with the IEMOCAP and Emo-DB databases were 78.90% and 85.73%, respectively. The RAVDESS database contains eight emotion categories, three of which, “calm”, “fear”, and “anger,” were identified with accuracies of 94.78%, 91.35%, and 84.60%, respectively, by the MLP. In contrast, the other five emotions were identified with less than 90.00% accuracy, as represented in Fig. 8. The MLP achieved an average accuracy with the SAVEE database of 75.38%. With the SAVEE database, “anger,” “neutral,” and “sad” were recognized with accuracies of 94.22%, 90.66%, and 85.33%, respectively, by the MLP classifier. IEMOCAP achieved an average accuracy of 84.00% with the SVM, while the MLP achieved an average accuracy of 80.23%. Figure 9 shows that the average accuracy achieved by the SVM with the IEMOCAP database is 84.00%.

Figure 7 Confusion matrix obtained by the SVM on the RAVDESS database for the SI experiment.

Figure 8 Confusion matrix obtained by the MLP on the RAVDESS database for the SI experiment.

Figure 9 Confusion matrix obtained by the SVM on the IEMOCAP database for the SI experiment.

Four publicly available databases are used to compare the proposed method. As illustrated in Table 8, the developed system outperformed (Guo et al., 2018; Chen et al., 2018; Meng et al., 2019; Özseven, 2019; Bhavan et al., 2019) on the Emo-DB dataset for the SD experiments. The OpenSMILE package was used to extract features in Özseven (2019). The accuracies obtained with the SAVEE and Emo-DB databases were 72% and 84%, respectively. In comparison to (Chen et al., 2018; Meng et al., 2019; Satt, Rozenberg & Hoory, 2017; Zhao et al., 2018), the proposed method performed well on the IEMOCAP database. The models in (Chen et al., 2018; Meng et al., 2019; Etienne et al., 2018) are computationally complex and require extensive periods of training. In the proposed method, AlexNet is used for the extraction process, and the FS technique is applied. The FS approach reduced the classifier’s workload while also improving efficiency. When using the RAVDESS database, the suggested technique outperforms (Zeng et al., 2019; Bhavan et al., 2019) in terms of accuracy.

Table 8 Comparison of the SD experiments with existing methods.

Database	Reference	Feature	Accuracy(%)	
RAVDESS	Bhavan et al. (2019)	Spectral centroids, MFCC and MFCC derivatives	75.69	
RAVDESS	Proposed approach	AlexNet + FS + RF	86.79	
RAVDESS	Proposed approach	AlexNet + FS + SVM	88.77	
SAVEE	Özseven (2019)	OpenSmile features	72.39	
SAVEE	Proposed approach	AlexNet + FS + RF	86.79	
SAVEE	Proposed approach	AlexNet + FS + SVM	88.77	
Emo-DB	Guo et al. (2018)	Amplitude spectrogram and phase information	91.78	
Emo-DB	Chen et al. (2018)	3-D ACRNN	82.82	
Emo-DB	Meng et al. (2019)	Dilated CNN + BiLSTM	90.78	
Emo-DB	Özseven (2019)	OpenSMILE features	84.62	
Emo-DB	Bhavan et al. (2019)	Spectral centroids, MFCC and MFCC derivatives	92.45	
Emo-DB	Proposed approach	AlexNet + FS + MLP	95.80	
Emo-DB	Proposed approach	AlexNet + FS + SVM	96.02	
IEMOCAP	Satt, Rozenberg & Hoory (2017)	3 Convolution layers + LSTM	68.00	
IEMOCAP	Chen et al. (2018)	3-D ACRNN	64.74	
IEMOCAP	Zhao et al. (2018)	Attention-BLSTM-FCN	64.00	
IEMOCAP	Etienne et al. (2018)	CNN + LSTM	64.50	
IEMOCAP	Meng et al. (2019)	Dilated CNN + BiLSTM	74.96	
IEMOCAP	Proposed approach	AlexNet + FS + MLP	89.12	
IEMOCAP	Proposed approach	AlexNet + FS + RF	86.23	

Table 9 illustrates that the suggested approach outperforms (Meng et al., 2019; Sun & Wen, 2017; Haider et al., 2021; Yi & Mak, 2019; Guo et al., 2019; Badshah et al., 2017; Mustaqeem, Sajjad & Kwon, 2020) for SI experiments using the Emo-DB database. The authors extracted low-level descriptor feature emotion identification and obtained accuracies with the Emo-DB database of 82.40%, 76.90%, and 83.74%, respectively, in Sun & Wen (2017), Haider et al. (2021), Yi & Mak (2019). Different deep learning methods were used for SER with the Emo-DB database in Meng et al. (2019), Guo et al. (2019), Badshah et al. (2017), Mustaqeem, Sajjad & Kwon (2020). In comparison to other speech emotion databases, the SAVEE database is relatively small. The purpose of using a pretrained approach is that it can be trained effectively with limited data. In comparison to (Sun & Wen, 2017; Haider et al., 2021), the suggested technique provides better accuracy with the SAVEE database. When using the IEMOCAP database, the proposed methodology outperforms (Yi & Mak, 2019; Guo et al., 2019; Xia & Liu, 2017; Daneshfar, Kabudian & Neekabadi, 2020; Mustaqeem, Sajjad & Kwon, 2020; Meng et al., 2019). The classification results of the proposed scheme show a significant improvement over current methods. With the RAVDESS database, the proposed approach achieved 73.50 percent accuracy. Our approach allowed us to identify multiple emotional states with Multiple languages with a higher classification accuracy while using a smaller model size and lower computational costs. In addition, our approach included a simple design and user-friendly operating characteristics, which can make it suitable for implementations such as monitoring people’s behavior.

Table 9 Comparison of SI experiments with existing methods.

Database	Reference	Feature	Accuracy(%)	
RAVDESS	Proposed approach	AlexNet + FS + MLP	82.75	
RAVDESS	Proposed approach	AlexNet + FS + SVM	80.94	
SAVEE	Sun & Wen (2017)	Ensemble soft-MarginSoftmax (EM-Softmax)	51.50	
SAVEE	Haider et al. (2021)	eGeMAPs and emobase	42.40	
SAVEE	Proposed approach	AlexNet + FS + MLP	75.38	
SAVEE	Proposed approach	AlexNet + FS + SVM	70.06	
Emo-DB	Badshah et al. (2017)	DCNN + DTPM	87.31	
Emo-DB	Sun & Wen (2017)	Ensemble soft-MarginSoftmax (EM-Softmax)	82.40	
Emo-DB	Yi & Mak (2019)	OpenSmile features + ADAN	83.74	
Emo-DB	Guo et al. (2019)	Statistical features and empirical features + KELM	84.49	
Emo-DB	Meng et al. (2019)	Dilated CNN + BiLSTM	85.39	
Emo-DB	Haider et al. (2021)	eGeMAPs and emobase	76.90	
Emo-DB	Lech et al. (2020)	AlexNet	82.00	
Emo-DB	Mustaqeem, Sajjad & Kwon (2020)	Radial basis function network( RBFN) + Deep BiLSTM	85.57	
Emo-DB	Proposed approach	AlexNet + FS + MLP	92.65	
Emo-DB	Proposed approach	AlexNet + FS + SVM	90.78	
IEMOCAP	Xia & Liu (2017)	SP + CNN	64.00	
IEMOCAP	Chen et al. (2018)	Dilated CNN + BiLSTM	69.32	
IEMOCAP	Guo et al. (2019)	Statistical features and empirical features + KELM	57.10	
IEMOCAP	Yi & Mak (2019)	OpenSmile features + ADAN	65.01	
IEMOCAP	Daneshfar, Kabudian & Neekabadi (2020)	IS10 + DBN	64.50	
IEMOCAP	Mustaqeem, Sajjad & Kwon (2020)	Radial basis function network( RBFN) + Deep BiLSTM	72.2	
IEMOCAP	Proposed approach	AlexNet + FS + MLP	89.12	
IEMOCAP	Proposed approach	AlexNet + FS + RF	86.23	

Conclusions and future work

In this research, the primary emphasis was on learning discriminative and important features from advanced emotional speech databases. Therefore, the main objective of the present research was advanced feature extraction using AlexNet. The proposed CFS approach explored the predictability of every feature. The results showed the superior performance of the proposed strategy with four datasets in both SD and SI experiments.

To analyze the classification performance of each emotional group, we display the results in the form of confusion matrices. The main benefit of applying the FS method is to reduce the abundance of features by selecting the most discriminative features and eliminating the poor features. We noticed that the pretrained AlexNet framework is very successful for feature extraction techniques that can be trained with a small number of labeled datasets. The performance in the experimental studies empowers us to explore the efficacy and impact of gender on speech signals. The proposed model is also useful for multilanguage databases for emotion classification.

In future studies, we will perform testing and training techniques using different language databases, which should be a useful evaluation of our suggested technique. We will test the proposed approach in the cloud and in an edge computing environment. We would like to evaluate different deep architectures to enhance the system’s performance when using spontaneous databases.

Supplemental Information

Supplemental Information 1 All codes used in this article in Python.

Click here for additional data file.

Additional Information and Declarations

Competing Interests

Author Contributions

Data Availability

The authors declare that they have no competing interests.

Ammar Amjad conceived and designed the experiments, performed the experiments, analyzed the data, performed the computation work, prepared figures and/or tables, authored or reviewed drafts of the paper, and approved the final draft.

Lal Khan performed the experiments, prepared figures and/or tables, authored or reviewed drafts of the paper, and approved the final draft.

Hsien-Tsung Chang conceived and designed the experiments, analyzed the data, authored or reviewed drafts of the paper, and approved the final draft.

The following information was supplied regarding data availability:

The codes are available in the Supplemental File.

The third party data is from the open access database RAVDESS and available at Livingstone, Steven R., & Russo, Frank A. (2018). The Ryerson Audio-Visual Database of Emotional Speech and Song (RAVDESS) [Data set]. In PLoS ONE (1.0.0, Vol. 13, Number 5, p. e0196391). Zenodo. https://doi.org/10.5281/zenodo.1188976.

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
