# Peer review of "Effect on speech emotion classification of a feature selection approach using a convolutional neural network"

_PeerJ Computer Science, doi:10.7717/peerj-cs.766_

## Round 0.1 · original submission · Major Revisions

The authors should revise the manuscript in light of both reviewers comments considering major revisions.

Reviewer 1 ·

Basic reporting

The paper is very well written however at several places poor sentence structure makes the paper difficult to read and create ambiguities. Professional proof reading is required before publication.

Experimental design

The contribution of the paper is the field is appreciable but the following shortcomings and must be addressed. .

In their proposed approach the authors have used the AlexNet for feature extractions. This network architecture is too old and has been superseded by several newer CNN based feed forward networks. Particularly the use of 11X11 convolutions in the AlexNet has been mostly discontinued and 3x3 kernels are used in most current networks.

The authors have used the transfer learning approach by using a pretrained AlexNet. However, they have not clearly mentioned the dataset used for pretraining of the Alexnet.

The authors do not specified the details of deep learning python libraries (e.g. Tensorflow/Keras or Pytorch) etc. in section 4.2.

In sub-section 5.0.1 the the number of neurons, activation functions and loss functions used to train the MLP classifier have not been specified. Similarly, hyperparameters used for the other methods are also not mentioned.

Validity of the findings

The findings have been evaluated by using the well known metrics in the field and conclusion is well stated.

·

Basic reporting

Language needs to be improved including the following particular issues:
Line 71: ‘have been founded’ should be ‘have been found’
Line 77-78: 'Therefore' opening the first two consecutive sentences
Line 109: ‘Emotions are categorized into two approaches’ should be ‘Emotions are categorized using two approaches’
Section 2.1 and literature review in general presents various kinds of techniques and their results without a chronological sequence

Experimental design

The method proposed in this paper extracts Alexnet based deep learning features from spectrogram for speech and then uses correlation based feature selection before feeding multiple shallow classifiers for emotion recognition. However the manuscript should address following two main issues:

1st issue:
A similar method is proposed by ‘Melissa N., et al 2017 and 2020’. They treat the speech spectrogram as an RGB image and uses Alexnet for feature extraction to classify emotions without the ‘feature selection step though.
The manuscript has not referred to these papers at all. The authors must highlight how the proposed method is different or better from the mentioned work.


Stolar, Melissa N., et al. "Real time speech emotion recognition using RGB image classification and transfer learning." 2017 11th International Conference on Signal Processing and Communication Systems (ICSPCS). IEEE, 2017.

Lech, Margaret, et al. "Real-time speech emotion recognition using a pre-trained image classification network: Effects of bandwidth reduction and companding." Frontiers in Computer Science 2 (2020): 14.

2nd issue:
The experimental setup for feature selection from speech features is not explained in detail. Both sections 3.4 and 3.5 provide the generic foundation of feature selection techniques without any explanation of implementation in the proposed setup.

Validity of the findings

The results section presents confusion matrices for different datasets without explaining their significance. The manuscript should justify the classification metrics analyzed in results in terms of their significance e.g. what are the consequences of classifying a particular emotion as another and what the confusion matrices aim to highlight.

Tables 3-6 present accuracies for different classifiers without mentioning ‘accuracy’ in the table tile or any axis

Line 398 in conclusion suggests that authors aim to test the results in cloud and edge computing environments. What is the rationale for such tests?

---

## Round 0.2 · Minor Revisions

The authors have not correctly and thoroughly revised the article based on previous comments. They should,
1. highlight their contributions with respect to Melissa N. Stolar; Margaret Lech 2017 and 2020
2. Add details of pipeline for extracting spectrogram feature
3. Address concerns related to validity of the findings

·

Basic reporting

NA

Experimental design

1. The authors now cited Melissa N. Stolar; Margaret Lech 2017 and 2020 which follow a similar approach to the proposed method however the similarities and differences in the proposed approach with these references in particular and other approaches, in general, must be highlighted.
.

2. The authors have explained in the response letter their rationale and strategy for feature selection from speech. However, it will be helpful for replication, etc. to add the implementation details in the manuscript itself about the pipeline for extracting spectrogram features and then selecting the most suitable.

Validity of the findings

As a response to the justification for classification metrics used in the results section e.g. the confusion matrix, the authors have referred to Table 3 which compares the nature of speech datasets. The suggestion here was to explain the reasons for using certain classification metrics/confusion matrices, what they indicate and what are the implications of the obtained results in real-world scenarios.

Additional comments

NA

---

## Round 0.3 · accepted · Accept

The authors have significantly revised the article in light of the 2nd revision comments.

---

## Author Rebuttal · Round 0.3

**Original Article Title:** "Effect on Speech Emotion Classification of a Feature Selection Approach Using a Convolutional Neural Network"

**Dear Editor:**

Thank you very much for giving us a chance to revise our manuscript entitled **Effect on Speech Emotion Classification of a Feature Selection Approach Using a Convolutional Neural Network**. We are very happy to have received a positive evaluation, and we would like to express our appreciation to you and all reviewers for the thoughtful comments and helpful suggestions. There are still three suggestions, which we have carefully considered and made every effort to address. We fundamentally agree with all the suggestions made by the reviewer and editor, and we have incorporated corresponding revisions into the manuscript. Our detailed, point-by-point responses to the editorial and reviewer comments are given below, whereas the corresponding revisions are marked in colored text in the manuscript file. Specifically, colored text indicates changes made in response to the suggestions of reviewers. Additionally, we have carefully revised the manuscript to ensure that the text is optimally phrased and free from typographical and grammatical errors. We believe that our manuscript has been considerably improved as a result of these revisions, and hope that our revised manuscript entitled **Effect on Speech Emotion Classification of a Feature Selection Approach Using a Convolutional Neural Network** is acceptable for publication in the Peerj Computer Science. We would like to thank you once again for your consideration of our work and for inviting us to submit the revised manuscript. We look forward to hearing from you.

**Best regards**
Hsien-Tsung Chang
Chang Chang Gung University
Department of Computer Science and Information Engineering Taoyuan, Taiwan
**E-mail:** smallpig@widelab.org

**Reviewer 1**
**Experimental design**
**(Concern#1)**
The authors now cited Melissa N. Stolar; Margaret Lech 2017 and 2020 which follow a similar approach to the proposed method however the similarities and differences in the proposed approach with these references in particular and other approaches, in general, must be highlighted`.
**Author response:** We thank the reviewer for the suggestion.
**Author action:** We updated the manuscript by highlighted the references with differences and similarities in the manuscript.
In existing literature (\cite{8085174, 4563453, 8270472, 10.3389/fcomp.2020.00014, BHAVAN2019104886, OZSEVEN2019320, Guo2018}), most of the studies used simulated databases with few emotional states. On the other hand, in the proposed study, we utilized eight emotional states for RAVDESS, seven emotional states for SAVEE, six emotional states for Emo-DB, and four emotional states for the IEMOCAP database. **Therefore, our results are state-of-the-art for both simulated and semi-natural databases** (page#7 Line#199).
In \cite {7883728}, a pre-trained DCNN model was introduced for speech emotion recognition. The outcomes were improved with seven emotional states (Emo-DB) **(simulated)** (page#5 Line#170). In \cite{4563453} the fully connected layer (FC7) of Alex Net was used for the extraction process. The results were evaluated on four different databases with **six emotions** (page#5 Line#180). In (\cite{8270472}), the existing approach used ALEXNet-SVM, experiments were performed on the EMO-DB database **(simulated)** with seven emotions (page#4 Line#156). An implementation of real-time voice emotion identification using AlexNet was described in (\cite{10.3389/fcomp.2020.00014}). When trained on the Berlin Emotional Speech (EMO-DB) **(simulated)** database with six emotional classes, the presented method obtained an average accuracy of 82\% (Page#7 Line#197,). Pretrained networks have many benefits, including the ability to reduce training time and improve accuracy. They also need fewer training data and deal directly with dynamic variables. **Our model results are based on multiple languages with multiple emotional states.**

**(Concern#2)**
The authors have explained in the response letter their rationale and strategy for feature selection from speech. However, it will be helpful for replication, etc. to add the implementation details in the manuscript itself about the pipeline for extracting spectrogram features and then selecting the most suitable.
**Author response:** We deeply appreciate the reviewer for his very insightful and constructive comments
**Author action:** We agree with the reviewer's assessment. Accordingly, throughout the manuscript, we have updated the manuscript by adding the detailed pipeline for extracting spectrogram features and also add the table of the number of most discriminative features after applied the CFS approach (page#7, Table#3(a),(b)). Figure 2 illustrates the pipeline for extracting features from CL4 (page#7 Table 3), the learning approach starts with an initial learning rate of 0.001 and gradually decreases with a drop rate of 0.1. By using 96 filters of 11x11x3, CL1 creates an array of activation maps. Consequently, CL4 generates 384 activation maps (3x3x192 filters). (page#9 Line271). We also updated figure 2 with captions (page#4, figure#2).
Because we used CFS(\cite{Wosiak2018}) approach for selecting the features, equation 5 represented a feature selection approach, where k represents the total number of features, $r_{cfi}$ represents the classification correlation of the features, and $r_{fifj}$ represents the correlation between features. The extracted features are fed into classification algorithms. CFS usually deletes

(backward selection) or adds (forward selection) one feature at a time. Table 3(b) gives the most discriminative number of selected features. (page#9 Line#283).

Validity of the findings
**(Concern#3)**
As a response to the justification for classification metrics used in the results section e.g. the confusion matrix, the authors have referred to Table 3 which compares the nature of speech datasets. The suggestion here was to explain the reasons for using certain classification metrics/confusion matrices, what they indicate and what are the implications of the obtained results in real-world scenarios.

**Author response:** We thank the reviewer for pointing out our mistake. Here, we apologize that we make a grave mistake.

**Author action:** A confusion matrix is an approach for describing the accuracy of the classification technique. For instance, if the data contains an imbalanced amount of samples in every group or more than two groups, the accuracy of the classification alone may be deceptive. Thus, calculating a confusion matrix provides a clearer understanding of what our classification model gets right and what kinds of mistakes it makes. We could noticed that confusion matrix were widely used for classification metrics in related researches(\cite{7956190,8421023,8873581}), we also cite those papers to make reader clarify why we used this metric. The row in the confusion matrix means the actual emotion classes in the confusion matrix, while the column indicates the predicted emotion classes. The results of the confusion matrix are used to evaluate the identification accuracy of the individual emotional labels. e.g, The Emo-DB database contains seven emotional categories, three of which, "sad", "disgust", and "neutral," were identified with accuracies of 98.88\%, 98.78\%, and 97.45\%, respectively, by the SVM illustrated in Figure 3. As shown in Figure 4, the SVM recognized "frustration" and "neutral" with the highest accuracies, 97.78\% and 92.45\%, with the SAVEE dataset. (page#12 Line#381).

The lightweight and most straightforward model presented in the proposed study has excellent accuracy. In addition, low-cost complexity can monitor real-time speech emotion recognition systems and show the ability for real-time applications (page#11 Line#340). Also, Our approach allowed us to identify multiple emotional states with Multiple languages with a higher classification accuracy while using a smaller model size and lower computational costs. In addition, our approach included a simple design and user-friendly operating characteristics, which can make it suitable for implementations such as monitoring people's behavior. (page#19 line#450).